# Contrasting Patterns of Fungal and Bacterial Endophytes Inhabiting Temperate Tree Leaves in Response to Thinning

**DOI:** 10.3390/jof10070470

**Published:** 2024-07-05

**Authors:** Beiping Liu, Chunhuan Li, Xiuhai Zhao, Chunyu Zhang, Xinyi He, Laiye Qu, Naili Zhang

**Affiliations:** 1State Key Laboratory of Efficient Production of Forest Resources, College of Forestry, Beijing Forestry University, Beijing 100083, China; liubeiping22@163.com (B.L.); lichunhuan318@163.com (C.L.); hexinyi0611@bjfu.edu.cn (X.H.); 2The Key Laboratory for Silviculture and Conservation of Ministry of Education, College of Forestry, Beijing Forestry University, Beijing 100083, China; 3Research Center of Forest Management Engineering of State Forestry and Grassland Administration, Beijing Forestry University, Beijing 100083, China; zhaoxh@bjfu.edu.cn (X.Z.); zcy_0520@163.com (C.Z.); 4Research Center for Eco-Environmental Sciences, Chinese Academy of Sciences, Beijing 100085, China; lyqu@rcees.ac.cn; 5Ecological Observation and Research Station of Heilongjiang Sanjiang Plain Wetlands, National Forestry and Grassland Administration, Shuangyashan 518000, China

**Keywords:** leaf damage, leaf endophytes, neighbor tree diversity, temperate forest, thinning intensity

## Abstract

The phyllosphere is an important but underestimated habitat for a variety of microorganisms, with limited knowledge about leaf endophytes as a crucial component of the phyllosphere microbiome. In this study, we investigated the mechanisms of communities and co-occurrence networks of leaf endophytes in response to forest thinning in a temperate forest. As we expected, contrasting responses of fungal and bacterial endophytes were observed. Specifically, the diversity of leaf endophytic fungi and the complexity of their co-occurrence networks increased significantly with thinning intensity, whereas the complexity of endophytic bacterial co-occurrence networks decreased. In particular, microbiota inhabiting damaged leaves seem to be more intensively interacting, showing an evident fungi–bacteria trade-off under forest thinning. In damaged leaves, besides the direct effects of thinning, thinning-induced changes in neighbor tree diversity indirectly altered the diversity of leaf fungal and bacterial endophytes via modifying leaf functional traits such as leaf dry matter content and specific leaf area. These findings provide new experimental evidence for the trade-offs between leaf endophytic fungi and bacteria under the different magnitudes of deforestation, highlighting their dependence on the presence or absence of leaf damage.

## 1. Introduction

Leaf microbial endophytes constitute a group of microorganisms inhabiting the leaves of plants, forming a harmonious symbiotic association with host trees after experiencing a long-term evolutionary history [1]. Intrinsic factors such as genotype, age, plant species, and developmental stage influence their diversity and composition [1,2], as well as external factors such as soil characteristics [3], the hydrothermal environment [4], neighboring plants [5], phytophagous insects [6], and plant pathogens [7]. The intricate environmental diversity enables leaf endophytic microbiota to demonstrate greater adaptability to environmental changes compared to inter-root microbes [8]. Despite the undeniable influence of leaf endophytic microbiota on plants [9], our understanding of the underlying mechanisms, particularly under varying thinning intensities, significantly lags behind the knowledge available for inter-root microbiota [10,11].

Thinning, the common anthropogenic disturbance being conducted prevalently in temperate forests, influences the structure and function of forest ecosystems by directly changing tree density and indirectly altering tree growth, functional traits, and composition through the modulation of sunlight penetration and precipitation [12,13,14,15]. The penetration of sunlight is closely associated with the traits and functions of leaves since they are photosynthetic organs [16]. The effects of thinning generally depend on the thinning intensity. Large-scale thinning usually leads to the excessive exploitation of forest resources, resulting in tree species loss, substantial variations in functional traits, and threats to ecosystem functions [17]. However, when managed appropriately, it has the potential to foster tree growth, promote leaf traits and their associations with microbial endophytes, alleviate competitive pressures among tree species [18], and improve the stability of forest communities [19,20].

The thinning-induced variations in tree spatial distribution play a crucial role in shaping ecological processes such as recruitment, growth, dispersal, and competition [21]. According to the “moderate disturbance hypothesis” proposed by Connell [22], when disturbance intensity is too high, it becomes challenging for the majority of species to survive, leading to the dominance of a small number of reproductively strong or expansive colonizers. Conversely, if disturbance intensity is too low, dominant competitors may become excessively strong and outcompete disadvantaged species, resulting in reduced species diversity [23,24]. Moderate thinning is deemed to stimulate the growth of specific leaf endophytic microbiota by increasing light availability, creating additional space resources, and enhancing the water use efficiency of leaves [25]. However, excessive thinning can decrease community-level and neighbor tree diversity, resulting in a significant change in the diversity and abundance of leaf endophytic microbiota inhabiting focal tree species [26,27]. Moreover, thinning-induced changes in neighbor tree richness may modulate herbivorous insects, conferring a cascading effect on leaf endophytic microbiota, a phenomenon often referred to as neighborhood effects [28]. The neighborhood effects can trigger changes in the herbivorous insect density and their feeding activity [29,30]. The extent of leaf damage by these insect herbivores has been predicted to modulate endophytic microbiota owing to the dispersal and colonization of some external microbes via the edges of damaged leaf tissues [27,29]. This raises an intriguing question: what are the differences in the environmental adaptation of endophytic microbiota residing in damaged leaves compared to those in healthy leaves?

In this study, we aim to investigate the responses of leaf endophytic fungal and bacterial assemblages to neighbor tree species richness and herbivorous insect-caused leaf damage in temperate forests under different thinning intensities. We established three plots with different thinning intensities (i.e., control check (CK), light, and moderate thinning) in a broad-leaved *Pinus koraiensis* forest in Northeast China. Five dominant tree species (i.e., *Pinus koraiensis*, *Ulmus davidiana* var. *japonica*, *Tilia amurensis*, *Fraxinus mandshurica*, and *Acer pictum* subsp. *mono*) were selected for the study. First, we hypothesized that thinning intensities can significantly change the diversity and composition of endophytic microflora in leaves, with a trade-off between leaf endophytic fungi and bacteria. Specifically, endophytic bacterial diversity may decrease as an enhancement of fungal diversity. Second, the intensities of thinning can significantly alter the network structure of leaf endophytes. Third, we predicted that leaf endophytes would be simultaneously regulated via the abundance of neighboring trees around the focal tree individual, via leaf functional characteristics, and via leaf damage caused by herbivorous insect feeding.

## 2. Materials and Methods

### 2.1. Study Site and Experimental Design

The study was carried out in a broad-leaved *Pinus koraiensis* forest at an altitude of 575~784 m, situated in the Jiaohe Forestry Experimental Area Administration of Jilin Province, China (127°45.287′–127°45.790′ E, 43°57.524′–43°58.042′ N). The average temperature from June to August here reaches 3.8 °C, and the average annual precipitation is 696 mm [31]. The soil type is mountain dark brown forest soil, with a depth range of 20 to 80 cm. The vegetation belongs to Changbai Mountain flora, with common tree species such as *Juglans mandshurica*, *Carpinus cordata*, *Quercus mongolica*, and so on. The undergrowth consists of *Syringa reticulata* subsp. *amurensis*, *Corylus mandshurica*, *Eleutherococcus senticosus*, etc. [32].

In December 2011, three 1ha plots were established in the broad-leaved *P. koraiensis* forest and were logged with different thinning intensities after obtaining the approval of the Jilin Provincial Forestry Department. The thinning intensities were as follows: 0% (control check, CK), 15% (light thinning, LT), and 30% (moderate thinning, MT) [33]. Five dominant species were selected, including *P. koraiensis* (PK), *Ulmus davidiana* var. *japonica* (UD), *Tilia amurensis* (TA), *Fraxinus mandshurica* (FM), and *Acer pictum* subsp. *mono* (AP). Four individual plants were randomly selected from each focal species in each plot for leaf sampling. To ensure consistency in sampling and minimize the effects of other environmental factors, we selected sample plots in forests with flat terrain and a slope of less than 5 degrees [34]. Additionally, the forest gaps around the target tree species were essentially uniform, thereby mitigating the potential influence of light and wind speed.

### 2.2. Sample Collection and Measurement

From late July to early August 2022, we collected leaves of the target tree species. Prior to this, the diameter at breast height (DBH) of each focal tree species was measured, and the neighboring trees surrounding each focal tree within a 5 m radius were recorded. Mature leaves were randomly selected from the mid-canopy layers, and their status was recorded as healthy or damaged. Leaves eaten by phytophagous insects were classified as damaged, while leaves without insect feeding were classified as healthy. The leaves were then put into sterile sealed bags and carried down the hill in a portable refrigerator. Vernier calipers were used to measure leaf thickness. The leaf thickness (LT) of each individual tree was obtained by averaging the thickness of five leaves, and all the leaves were scanned for further analysis. Image Pro Plus was used to determine the area (LA), length (LL), and width (LW) of the blade.

In the meantime, three leaves were chosen at random for each sample, and their fresh weight was determined. The leaves were then degreened at 105 °C for 15 min and dried at 65 °C to a constant weight for dry weight determination. The leaf dry matter weight (LDMC) was determined as the ratio of leaf dry weight to leaf fresh weight, and the specific leaf area (SLA) was calculated as the ratio of leaf area to leaf dry matter weight. The measurements of leaf trait variables per individual tree were taken three times, and the mean values were recorded. After grinding the leaves for 150 s with a 55 Hz grinding instrument, the nutrient content of the leaves was measured after passing through a 150-mesh screen. An elemental analyzer was used to measure the total nitrogen (N) and total carbon (C) content of the leaves (Vario EL Cube, Elementar, Germany) and to obtain the C/N ratio of the leaves. An inductively coupled plasma mass spectrometer was used to determine the content of P, K, Ca, Cu, Mg, and Zn in leaves (ICP-MS, NexION 300X, Perkin Elmre, Shelton, CT, USA), and the leaves were dissolved via the HNO_3_/HClO_4_ method.

Five soil cores beneath the canopy of each target tree were collected and thoroughly mixed into a composite sample, which was then placed in sterile sealed bags and transported to a lab for analysis. After removing impurities with a 2 mm sieve, the remaining soil was measured for soil moisture and pH value. Ten grams of fresh soil was weighed and dried at 105 °C to a constant mass. The dried soil was then reweighed to determine its moisture content (SWC). Soil pH was measured using a pH meter (PB-10, Sartorius AG 21992271, Göttingen, Germany) with a water-to-soil volume ratio of 1:2.5.

### 2.3. DNA Extraction, PCR Amplification, and Sequencing of Leaf Endophytic Bacterial and Fungal Communities

The subsamples of leaves were washed with running water, soaked in 75% alcohol for 1 min, soaked in pure water for another 1 min, and, finally, washed with 75% alcohol again for surface sterilization. After allowing all the alcohol to evaporate, the leaves were placed in aseptic bags and stored at −80 °C to measure endophytic fungi and bacteria in the leaves.

DNA extraction was carried out following the instructions of the TruSeqTM DNA Sample Prep Kit. NanoDrop 2000 and 1% agarose gel electrophoresis were utilized to analyze the concentration, purity, and quality of DNA extracted from the leaves. The extracted leaf DNA served as the template for polymerase chain reaction (PCR) amplification. The primers, 799F-AACMGGATTAGATACCCKG and 1193R-ACGTCATCCCCACCTTCC, were used to amplify endophytic bacterial variable region V3–V4 [35]. The PCR amplification process consisted of 3 min of predenaturation at 95 °C and 27 cycles (denatured at 95 °C for 30 s, annealed at 55 °C for 30 s, and extended at 72 °C for 30 s). Finally, the reaction time was extended for 10 min at 72 °C and then stored at 4 °C. The primers, fITS7F-GTGARTCATCGAATCTTTG and ITS4R-TCCTCCGCTTATTGATATGC, were used to amplify the endophytic fungal ITS2 region [36]. The process of PCR amplification consisted of predenaturation at 95 °C for 5 min, in 30 cycles (denaturated at 94 °C for 60 s, annealed at 58 °C for 50 s, and extended at 68 °C for 60 s), and finally an extension at 68 °C for 10 min and preservation at 10 °C. The PCR amplification system consisted of a 20 μL mixture containing 4 μL of 5 × Fast Pfu buffer, 2 μL 2.5 mM dNTPs, 0.8 μL of each primer (5 μM), 0.8 μL of Fast Pfu polymerase, 10 ng of the DNA template, and ddH2O. The AxyPrep DNA gel extraction kit (Axygen Biosciences, Union City, CA, USA) was utilized for purifying PCR products and detecting them with 2% agarose gel electrophoresis. The Illumina MiSeq platform (Illumina, San Diego, CA, USA) was used to sequence the purified PCR products following the established protocol of Megi Biopharmaceutical Co., LTD. (Shanghai, China). The original sequencing data were stored in FASTQ format.

### 2.4. Bioinformatics Analysis

We selected Quantitative Insights into Microbial Ecology 2 (QIIME2) (v 2022.08) [37], an advanced microbiome analysis platform, for the detailed processing of all raw FASTQ data. First, the data import process was completed manually by executing the “qiime tools import” command in QIIME2. During the data cleaning and preprocessing phase, we used the “qiime dada2 denoise paired” plug-in [38], which incorporates advanced functions including quality pruning, noise reduction, sequence merging, and chimera detection. The approach efficiently grouped non-singleton amplified sequence variants (ASVs).

After quality filtering, the ASV abundance table was normalized to achieve the consistency of sampling depth using the “Interactive Sample Detail” feature within the “table. qzv” file. The “qiime feature-classifier classify-sklearn” plugin in QIIME2 was then employed to classify fungal and bacterial ASVs. The ASVs were merged and taxonomically classified by a trained Naive classifier based on the Greengenes database, clustering species with 99% similarity. Two authoritative reference databases, SILVA (www.arb-silva.de/act (accessed on 24 May 2024)) and UNITE (https://unite.ut.ee/ (accessed on 24 May 2024)), were used for the taxonomic assignment of endophytic fungi and bacteria, respectively. The resulting endophytic fungi amplicon sequences were validated via comparison with the NCBI database.

### 2.5. Statistical Analysis

All statistical analyses and data visualizations in this study were performed using R statistical software (v.4.2.3). We conducted tests to evaluate the effects of different thinning intensities, tree species, damage, and their interactions on the assembly of endophytic bacteria and fungi. The lmer function from the lme4 package was employed to fit a mixed linear model, with the thinning intensity, tree species, and damage treated as fixed effects, and the DBH of target tree species and the composition of surrounding tree species treated as random effects. The significance of the fixed effects was assessed using the Anova function from the Car package. To examine the diversity and composition of endophytic fungi and bacteria, leaf traits, soil variables, and diversity of neighboring trees under different thinning intensities, we performed the Kruskal–Wallis nonparametric test. Differences between different thinning intensities were further analyzed using the Wilcoxon test (Wilcox.test function).

To identify the bacterial genera exhibiting significant differences in relative abundance across various thinning intensities, an LEfSe analysis was employed, with a threshold LDA > 2 and *p* < 0.05, and the bacterial genera with relative abundance greater than 0.05% were screened. A correlation network analysis was performed using the igraph package, with species selected based on Spearman correlation |r| > 0.6, *p* < 0.05. The ggclust package [39] was utilized to analyze core species within the network, categorizing node attributes into four types based on their topological characteristics. The fungal community function was predicted using the FUNGuild method [40], while the bacterial community function was predicted using the FAPROTAX method [41]. The structural equation modeling (SEM) posits a causal pathway where thinning intensity affects the richness of endophytic bacteria and fungi in leaves through altering neighbor tree diversity, soil variables, and leaf traits, as conceptualized (Appendix A); this pathway was constructed using the piecewiseSEM package [42]. The path coefficient expresses the direction and intensities of the direct influence between the two variables. Graphs were generated using the ggplot2 package.

## 3. Results

### 3.1. Alpha Diversity of Leaf Endophytic Fungi and Bacteria

The mixed-effect model analysis revealed that the alpha diversity (i.e., richness and Shannon index) of the leaf endophytic bacteria was significantly influenced by tree species, the degree of damage, and their interaction. However, thinning intensity and tree species identity had significant effects on the alpha diversity of endophytic fungi (Table 1). Thinning intensity notably increased the alpha diversity of endophytic fungi. It decreased that of the endophytic bacterial community, compared to the CK plots (Figure 1). The richness and Shannon index of the leaf endophytic bacterial community were lowest in LT plots, whereas those of leaf endophytic fungi were highest in LT plots.

In *F. mandshurica*, the endophytic bacterial richness and Shannon index were significantly lower in MT plots than in CK plots (Appendix A). In *U. davidiana*, the richness of endophytic bacteria was significantly lower in MT plots than in CK plots, while the richness of endophytic fungi showed the opposite trend (Appendix A). Except for *T. amurensis* species, the diversity of leaf endophytic bacteria, indicated by both richness and the Shannon index, was lower in thinning plots compared to CK plots, with diversity generally higher in damaged leaves than in healthy leaves (Appendix A). Conversely, the diversity of leaf endophytic fungal, except for in *T. amurensis*, increased due to forest thinning, with diversity generally lower in damaged leaves than in healthy leaves (Appendix A).

### 3.2. Composition of Leaf Endophytic Fungi and Bacteria

We analyzed 120 leaf samples to identify fungal and bacterial endophytes, resulting in the identification of 4672 bacterial ASVs and 6390 fungal ASVs. The top 10 genera with the highest relative abundance of endophytic bacteria at the genus level included *Achromobacter*, *Pseudomonas*, *Sphingomonas*, *Rhodococcus*, *Delftia*, *Methylobacterium*, *Agrobacterium*, *Friedmanniella*, *Variovorax*, and *Curtobacterium* (Figure 2). The relative abundance of the remaining genera and unknown species accounted for more than 25% across all plots. Regarding fungal endophytes, the top 10 genera with the highest relative abundance were *Tripospermum*, *Vishniacozyma*, *Sphaerulina*, *Dioszegia*, *Trichomerium*, *Pucciniastrum*, *Buckleyzyma*, *Cyphellophora*, *Papiliotrema*, and *Taphrina* (Figure 2). More than 50% of the remaining genera and unknown species across all plots were also identified.

We observed that thinning intensity influenced the relative abundance of endophyte taxa at genus levels (Figure 3). Among the endophytic bacteria, genera such as *Sphingomonas*, *Variovorax*, and *Devosia* exhibited the highest relative abundance in CK plots (Figure 3). Conversely, the relative abundance of *Pseudomonas* and *Rhodococcus* increased with the thinning intensity. For endophytic fungi, genera such as *Tripospermum*, *Buckleyzyma*, *Cyphellophora*, *Apiotrichum*, and *Neoacrodontiella* decreased in relative abundance with increasing thinning intensity (Figure 3). Notably, among endophytic fungi, *Camptophora*, *Tricladiella*, and *Genolevuria* had the highest relative abundance in LT plots.

*F. mandshurica*, the relative abundance of endophytic bacteria, including *Pseudomonas*, *Sphingomonas*, *Variovorax*, and *Devosia*, significantly changed with thinning intensity (Appendix A). In *U. davidiana*, the relative abundance of *Sphingomonas* and *Variovorax* significantly decreased with thinning intensity (Appendix A). Regarding endophytic fungi, in *T. amurensis*, the relative abundance of endophytic fungi such as *Buckleyzyma*, *Cyphellophora*, and *Neoacrodontiella* decreased with thinning intensity, which was also true for *Tripospermum*, *Cyphellophora* in *P. Koraiensis*, and for *Buckleyzyma* in *U. davidiana* (Appendix A).

In the damaged leaves, only three genera of endophytic bacteria, namely *Sphingomonas*, *Rhodococcus*, and *Variovorax*, were significantly impacted by thinning intensity (Appendix A). In *U. davidiana*, the relative abundance of the endophytic fungi, particularly *Neoacrodontiella*, significantly responded to thinning intensity in both healthy and damaged leaves, with the highest abundance observed in CK plots (Appendix A). Discrepancies in the abundance of certain fungi between healthy and damaged leaves in response to thinning intensity were also observed in *A. pictum*, *P. Koraiensis*, and *T. amurensis* (Appendix A).

### 3.3. Co-Occurrence Networks of Leaf Endophytic Fungi and Bacteria

A co-occurrence network was performed using the Spearman correlation coefficient. Node attributes were compared in Table 2, including the number of nodes, edges, edge density, average degree, and relative modularity. For the endophytic bacterial community, the CK plot exhibited a higher number of nodes, edges, edge density, average degree, and relative modularity compared to LT and MT plots. The network structure of the bacterial community appeared more complex in the CK plot, which was reduced with the thinning intensity. For the endophytic fungal community, however, the MT plot showed a higher number of nodes, edges, edge density, average degree, and relative modularity compared to CK and LT plots. Increased thinning intensity resulted in greater complexity in the structure of the endophytic fungal community network.

The network nodes were categorized by genus, with *Sphingomonas* showing the highest proportion among endophytic bacteria and *Dioszegia* among endophytic fungi (Figure 4a). The key taxa were classified at the genus level via the core node discrimination method of the Zi-Pi network (Figure 4b). The topological characteristics of nodes divide node attributes into four types: module hubs (nodes with high connectivity within modules), connectors (nodes with high connectivity between two modules), network hubs (nodes that have high connectivity throughout the network), and peripherals (nodes that do not have high connectivity within or between modules). The remaining three types of nodes, excluding peripherals, are generally classified as key species. After the modularization of the endophytic bacterial network, the intra-module connectivity (Zi) and inter-module connectivity (Pi) of the different nodes were calculated, showing 14 key species in the CK plots, 7 key species in the LT plots, and 12 key species in the MT plots (Figure 4b). These key species act as modular hubs and connectors, for instance, ASV_3211 appeared in both CK and MT plots, but only the Cystobacterace family has been identified in ASV_3211. In addition, CK and MT samples shared the presence of the *Methylbacterium* genus. In endophytic fungi, the Zi-Pi values indicated 8 key species in CK plots, 5 key species in LT plots, and 13 key species in MT plots, with these key species functioning as modular hubs. Both CK and LT plots contained ASV_2271, identified as a Chaetothyriales order, and the CK and LT plots shared a common genus of *Erythrobasidiomycetes* (Figure 4b). We also observed that the leaves of *T. amurensis*, *A. pictum*, *U. davidiana*, and *P. Koraiensis* harbored key species belonging to the genera *Methylobacterium* and *Sphingomonas* within their endophytic bacterial communities. Similarly, the endophytic fungi in the leaves of *T. amurensis*, *A. pictum*, and *P. Koraiensis* featured key species belonging to the *Erythrobasidium* genus (Appendix A).

### 3.4. Functional Prediction of Leaf Endophytic Fungi and Bacteria

Using the FAPROTAX functional annotation method, the ecological categories of endophytic bacteria were classified into three types, i.e., those involved in the carbon cycle, nitrogen cycle, and energy source. The dominant communities were Chemoheterotrophy, Aerobic Chemoheterotrophy, and Anaerobic Chemoheterotrophy, collectively accounting for over 50% (Figure 5a). As thinning intensity increased, functional groups involved in nitrogen respiration and nitrate reduction significantly increased. Concurrently, the functional groups of Aerobic Chemoheterotrophy and Chemoheterotrophy significantly decreased (Figure 5b).

The ecological classification of endophytic fungi is based on FUNGuild functional prediction, divided primarily into three types, i.e., saprotroph, pathotroph, and symbiotroph. The dominant communities include litter saprotrophs, plant pathogens, and soil saprotrophs, collectively accounting for more than 50% of the communities (Figure 5a). Litter saprotrophs declined significantly with the increase in thinning intensity, whereas plant pathogens increased significantly under the same condition. However, there was no significant difference in soil saprotrophs (Figure 5b).

Except for *U. davidiana* species, the endophytic bacteria involved in Aerobic Chemoheterotrophy and Chemoheterotrophy decreased with thinning intensity. In *F. mandshurica*, the endophytic bacteria involved in nitrogen respiration and nitrate reduction increased with thinning intensity (Appendix A). Similarly, the plant fungal pathogen also increased with thinning intensity (Appendix A, lower panels). In *P. Koraiensis*, the litter saprotroph of endophytic fungi decreased with thinning intensity.

There were no significant differences in endophytic bacteria in both healthy and damaged leaves (Appendix A). In *F. mandshurica*, among the endophytic fungi of damaged leaves, the plant pathogen was the highest in the LT plot. In contrast, in healthy leaves, the endophytic fungi increased with the rise in thinning intensity, peaking in the MT plot (Appendix A, lower panels). In *P. Koraiensis*, litter saprotrophs of endophytic fungi decreased with the increasing thinning intensity, observed in both healthy and damaged leaves.

### 3.5. Drivers Triggering Variations in the Assemblages of Leaf Endophytic Fungi and Bacteria

Thinning intensity had significant effects on SLA, LDMC, Zn, and P (Table 3). Using stepwise forward regression, we selected leaf traits that significantly explained the richness of leaf bacterial and fungal endophytes (Appendix A). A preliminary model was constructed based on previous studies (Appendix A). During the adjustment process of the structural equation model, SLA and LDMC were further identified as key leaf trait factors regulating the relationship between thinning intensity and leaf endophytic bacterial communities. Therefore, in the structural equation model, the composite variables of leaf traits comprised SLA and LDMC, and pH and soil moisture content were categorized as soil variables (Figure 6).

In healthy leaves, thinning intensity positively influenced the richness of endophytic fungi and bacteria, although the effect was not statistically significant (Figure 6). Moreover, thinning intensity indirectly affected the richness of the leaf endophytic fungi through modulating leaf traits (Figure 6b). Additionally, the richness of endophytic bacteria was significantly affected by leaf traits, as well as soil variables (Figure 6a). However, soil variables such as soil pH and water content did not affect endophytic fungal richness.

In damaged leaves, thinning intensity significantly negatively affected the richness of endophytic bacteria, while it significantly positively influenced the richness of endophytic fungi (Figure 6c,d). Furthermore, thinning intensity conferred indirect effects on the richness of both endophytic bacteria and fungi by modulating the neighbor tree diversity, which consequently changed the leaf traits of focal tree species and their associated bacterial and fungal richness (Figure 6c,d). Additionally, in damaged leaves, soil variables were found to positively affect leaf traits (Figure 6c).

## 4. Discussion

### 4.1. The Contrasting Responses of the Diversity of Leaf Endophytic Fungi and Bacteria to Thinning Intensity

The susceptibility of soil bacteria to abiotic stresses is higher compared to fungi [43,44]. However, there is limited literature regarding leaf endophytes in this context. Our findings indicate that under light thinning intensity, leaf endophytic fungal diversity significantly surpassed that of the control group, while leaf endophytic bacterial diversity was slightly lower in the thinning treatment compared to the control, although not significantly different (Figure 1). This suggests that leaf endophytic fungi exhibit greater sensitivity to thinning, whereas leaf endophytic bacteria demonstrate greater stability in response to thinning. Therefore, temperate forest thinning may intensify antagonistic interactions between leaf endophytic fungi and bacteria. In general, soil bacteria and fungi compete for resources, resulting in antagonism between these microbial groups [45], which may provide a plausible explanation for the contrasting trends observed in the diversity of fungal and bacterial communities. Furthermore, we observed that the most significant increase under the thinning gradient was in pathogenic fungi, while the dominant decrease in bacterial community diversity was observed in chemo-energy heterotrophs (Figure 5), supporting the notion that changes in bacterial diversity were driven by competition. Additionally, we found that host plant identity had a more significant impact on the structure of leaf endophytic bacterial communities compared to thinning intensity (Appendix A, Table 1). This finding aligns with previous studies, indicating that the host plant primarily regulates leaf endophytes and is less influenced by inter-host dispersal [26], which helps explain the lack of significant changes in leaf endophytic bacterial diversity with thinned intensity.

In terms of community composition, the temperate forest thinning significantly increased the relative abundance of *Rhodococcus* spp. in the endophytic bacterial community (Figure 2 and Figure 3) and concurrently reduced the relative abundance of *Cyphellophora* spp. in the endophytic fungal community. Conversely, thinning led to a significant increase in the relative abundance of *Camptophora* spp. in the endophytic fungal community (Figure 2 and Figure 3). Thinning is generally predicted to alter the community structure of host trees, accelerating nutrient availability and modifying organic matter content [46]. Moreover, thinning can cause significant changes in habitat conditions, such as increased exposure to direct sunlight and soil disturbance [47]. The changes can greatly influence the makeup of microbial communities, resulting in the emergence of communities adapted to the new habitat conditions and resource utilization strategies. Consequently, leaf endophytes with adaptive advantages to the altered habitat conditions and resource utilization strategies, such as the fungal and bacterial species mentioned above, become more prevalent, while non-adapted communities may be outcompeted and eliminated through competition.

### 4.2. The Contrasting Trends of the Complexity of Leaf Endophytic Fungal and Bacterial Co-Occurrence Networks

Thinning can exert significant effects on plant leaves by altering canopy density [48], water content [49], and the diversity of neighboring species [50]. We observed a progressive reduction in the complexity of the co-occurrence network of leaf bacterial endophytes with increasing thinning intensity, resulting in a simpler network topology, which is consistent with changes in diversity (Figure 4). This indicates that thinning leads to a decrease in interactions among leaf endophytic bacteria. Conversely, the topological features of the network of leaf endophytic fungi became progressively more complex with increasing thinning intensity (Figure 4). These characteristics reflect the level of connectivity and interactions among microorganisms [51]. One explanation for this topological variation is the differentiation of ecological niches within leaf tissues due to changes in the diversity of leaf endophyte communities triggered by thinning intensity [52]. The increased diversity of leaf endophytic fungi associated with higher thinning intensity led to a more homogeneous distribution of ecotopes for leaf endophytic fungi. Weak ecotope differentiation may enhance interactions between microorganisms [53]. Conversely, the decrease in the diversity of leaf endophytic bacteria may have resulted in significant ecotope differentiation, reducing competition, and allowing microorganisms to coexist within the community for extended periods. Additionally, this ecological niche differentiation may have inhibited interactions between different species of endophytic bacteria [54]. Similarly, bacteria and fungi were predominantly positively correlated in microbial networks under thinning. Positive and negative correlations in microbial networks represent mutually beneficial symbiotic or competitive relationships among interrelated species [55]. This pattern may arise from resource abundance and reduced competition among bacterial species, resulting in higher synergism between bacterial communities [56].

Node attributes in microbial networks can be categorized into four types based on the node’s topological characteristics: module hubs, connectors, network hubs, and peripherals. Typically, network hubs denote nodes that have a high degree of connectivity throughout the network (Zi > 2.5 and Pi > 0.62), i.e., nodes that play an important role both within modules and between different modules [57]. From an ecological perspective, peripherals are considered specialists, while network hubs are super-generalists. Due to environmental determinism, specialists constitute a major part of most systems, whereas super-generalists are very few [51,58]. In this study, network hubs were present neither in bacterial nor fungal networks. However, module hubs and connectors, which are closer to generalists, were found to varying degrees. These module hubs and connectors play an important role, highlighting their significance in bacterial and fungal endophytic network structures in temperate forests.

Given the intricate nature of leaf microbial communities, the concepts of microbial persistence (core taxa) and their significance in the microbial network (pivotal taxa) have been utilized to identify microbes playing key roles in leaf communities [59,60]. Furthermore, as the thinning intensity changed, different ASVs were assigned to different modules, indicating that thinning intensity altered the key taxa within the network of the leaf endophyte community. Most of the key taxa showed strong specificity for leaf endophytes. Notably, the CK plots and MT plots shared a genus, *Methylobacterium*. It has been well documented that the enrichment of plant biomes with the genus *Methylobacterium* can promote plant growth and productivity, as well as plant drought tolerance [61,62]. For leaf endophytic fungi, the CK and LT plots share a genus, *Erythrobasidium*, whose members have not been well studied, with only eight species found in nature so far [63,64], and whose functions are even less understood. It should be noted, however, that these key taxa are not the same across time and environments, that is, key species may play a critical role only under certain conditions [58,65]. Our results support that this environmental deterministic influence applies equally well in leaf endophyte communities.

For different tree species, we found that there was a high variability of bacterial and fungal key taxa among the different tree species, suggesting that the key taxa of the endophytic bacteria in leaves are highly plant species-specific. Interestingly, when extending the bacterial key taxa to the genus level, we found the genus *Sphingomonas* existed in four of the tree species. Bacteria of the genus *Sphingomonas* are widespread in nature due to their low nutrient concentration requirements and have been shown to have resistance to bacterial pathogens. Additionally, they can spread vertically between plant generations through seeds [66]. Overall, when investigating the network complexity of leaf endophytes, the key taxa should be well considered.

### 4.3. Leaf Trait-Mediated Thinning Effects on Leaf Fungal and Bacterial Endophytes

Leaf surfaces present hostile environments where inhabiting microorganisms are exposed to various physicochemical stresses, including intense light and ultraviolet (UV) radiation, fluctuating temperature and humidity, and limited nutrient availability [27]. Our findings indicate that both endophytic bacterial and fungal community richness were significantly positively influenced by leaf functional traits (Figure 6). Previous studies demonstrate that changes in endophytic microbial colonization and diversity are regulated by leaf economic profiles, which encompass trade-off strategies for leaf nutrient utilization [67]. It is because most leaf endophytes are horizontally transmitted via airborne spores, that endophyte spores and mycelium inevitably interact with leaf traits. Upon reaching the leaf surface, endophytes typically colonize the tissue through wounding and stomatal infiltration [15]. Consequently, traits related to leaf defense performance, such as specific leaf area and leaf dry matter mass, become particularly important for leaf endophytic bacterial communities. Furthermore, we found that leaf endophyte communities were less influenced by neighbors (Figure 6). This further corroborates our previous conclusion that endophytes may be primarily regulated by the host and less influenced by inter-host dispersal.

Unlike inter-root and soil microbial communities, leaf microbes are inevitably influenced by pollinating insects and herbivorous visitors, which introduce and redistribute microbiota [30]. Our findings reveal that in healthy leaves, soil properties can positively influence leaf endophytic bacterial abundance, whereas this relationship was not significant among damaged leaves. Previous studies indicate that the bacterial composition in leaves exhibits greater similarity to taxa present in soil than to bacteria in flowers and fruits [68]. This suggests that bacterial members in the interleaf microflora in the healthy statues are primarily soil-derived, emphasizing the essential role of soil properties in shaping the bacterial community. Furthermore, we found that neighbor diversity, in the presence of leaf damage, indirectly affects leaf endophyte communities by influencing leaf functional traits (Figure 6). Functional traits, serving as the foundation of leaf economic profiles, effectively express the ecological strategies of plants in resource acquisition and the optimal allocation of limited available resources [69,70]. The intensification of thinning is expected to decrease the neighborhood diversity of the target tree species and further increase ecological niches in the community, leading to an adjustment of the ecological strategy of the target tree species from a defensive to an aggressive development strategy. Finally, our results further elucidate that the effects of thinning on leaf endophytes may be primarily mediated by leaf damage, indicating that airborne microbial transmission and secretions from insect feeding possibly contribute to the reconstitution of the leaf microbial community.

Unlike those inhabiting healthy leaves, thinning had significant negative and positive effects on leaf endophytic bacteria and endophytic fungi in damaged leaves, respectively (Figure 6). Compared to the plant surface, the inner layers of leaves may provide a richer nutrient environment that protects microorganisms from external atmospheric fluctuations, including UV radiation and moisture [27]. However, this proximity to the plant’s immune surveillance mechanisms and defense compounds may inhibit the colonization potential of endophytes, and leaf damage provides a pathway for changes in the microbial community. Damaged leaves expose the living environment for endophytes, triggering the growth of endophytic pathogenic fungi, consistent with previous observations [71,72] that many pathogens causing forest leaf diseases require wounds or scrapes to penetrate host tissues, typically caused by mechanical damage from herbivores or other physical factors such as wind or water [73]. In contrast, endophytic bacteria may be inhibited by fungal growth during competition, leading to a decrease under thinning gradients, representing a trade-off between different thinning intensities on leaf endophyte communities. The competition of both bacteria and fungi for resources among limited ecological niches inevitably results in a tendency for them to outcompete each other [54]. This further explains that the changes in leaf endophyte communities’ diversity and co-occurrence network construction under different thinning intensities may be mediated by leaf damage.

## 5. Conclusions

Leaf endophytic fungi and bacteria exhibited contrasting responses to thinning intensity in temperate forest ecosystems. Specifically, endophytic fungal diversity increased significantly with thinning intensity, accompanied by an increase in the complexity of the co-occurrence network. Conversely, bacterial diversity did not show significant differences, but the complexity of the symbiotic network decreased as the thinning intensity increased. Moreover, the contrasting effects of thinning intensity on leaf fungal and bacterial endophytes were particularly noticeable in damaged leaves, compared to in healthy leaves, suggesting that the condition of the leaves—whether damaged or healthy—may be critical for constructing the trade-offs between leaf fungal and bacterial endophytes under forest thinning. Additionally, leaf endophytes were more strongly modulated by their host tree, with leaf functional traits having a significant positive effect on them, and the neighbor tree diversity also conferred an indirect effect on leaf endophytes via altering the leaf traits of host trees. Overall, our findings provide insight into the trade-offs between fungal and bacterial endophytes in response to forest thinning, particularly in damaged leaves.

## Figures and Tables

**Figure 1 jof-10-00470-f001:**
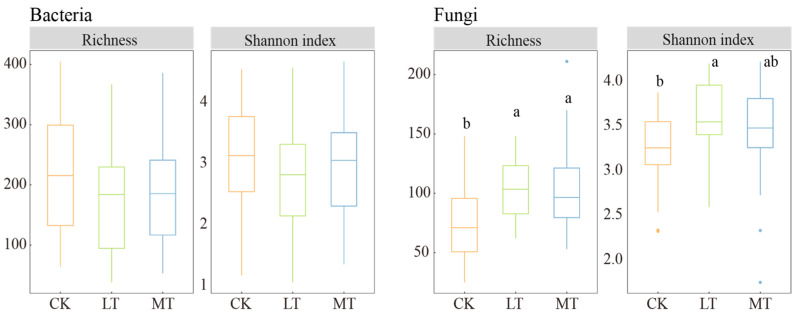
Alpha diversity index of endophytic fungi and bacteria under different thinning intensities. CK: control check; LT: light thinning; MT: moderate thinning. Different letters indicate significant differences (*p* < 0.05). Asterisks are outliers.

**Figure 2 jof-10-00470-f002:**
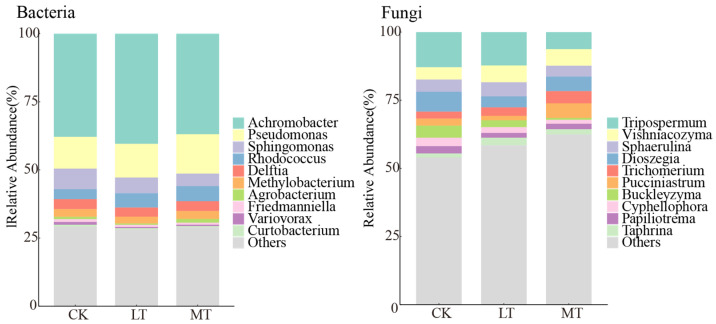
Composition of endophytic fungal and bacterial communities at different thinning intensities. CK: control check; LT: light thinning; MT: moderate thinning.

**Figure 3 jof-10-00470-f003:**
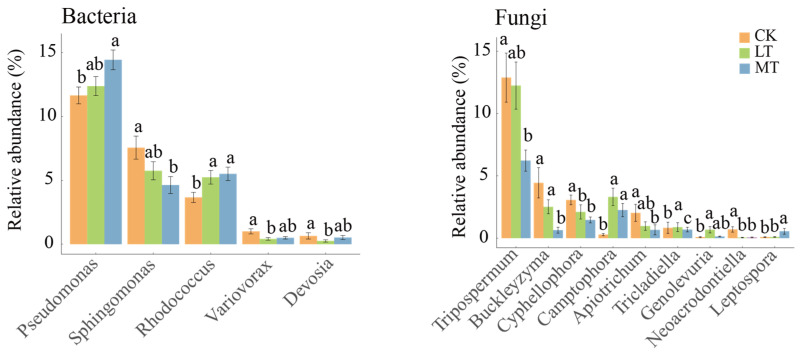
Relative abundance of endophytic bacterial and fungal communities at different thinning intensities. CK: control check; LT: light thinning; MT: moderate thinning. Values are means ± SE for panels (*n* = 40). Different letters represent significant differences (*p* < 0.05).

**Figure 4 jof-10-00470-f004:**
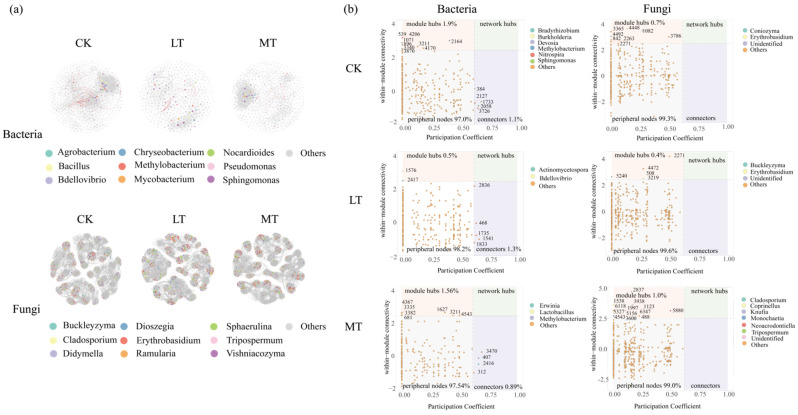
Network structure of endophytic fungal and bacterial communities across different thinning intensities. (**a**) Co-occurrence networks of endophytic bacterial and fungal communities. Gray lines indicate positive correlations and red lines indicate negative correlations. The nodes are colored based on the genus-level taxonomic classification of microbes. The node size corresponds to the degree level. (**b**) Zi-Pi diagram of the ecological network of endophytic bacterial and fungal communities. According to corresponding criteria, we identified module hubs (Zi ≥ 2.5, Pi < 0.62), connectors (Zi < 2.5, Pi ≥ 0.62), and network hubs (Zi ≥ 2.5, Pi ≥ 0.62), which are referred to as keystone nodes. All other nodes were categorized as peripherals. CK: control check; LT: light thinning; MT: moderate thinning.

**Figure 5 jof-10-00470-f005:**
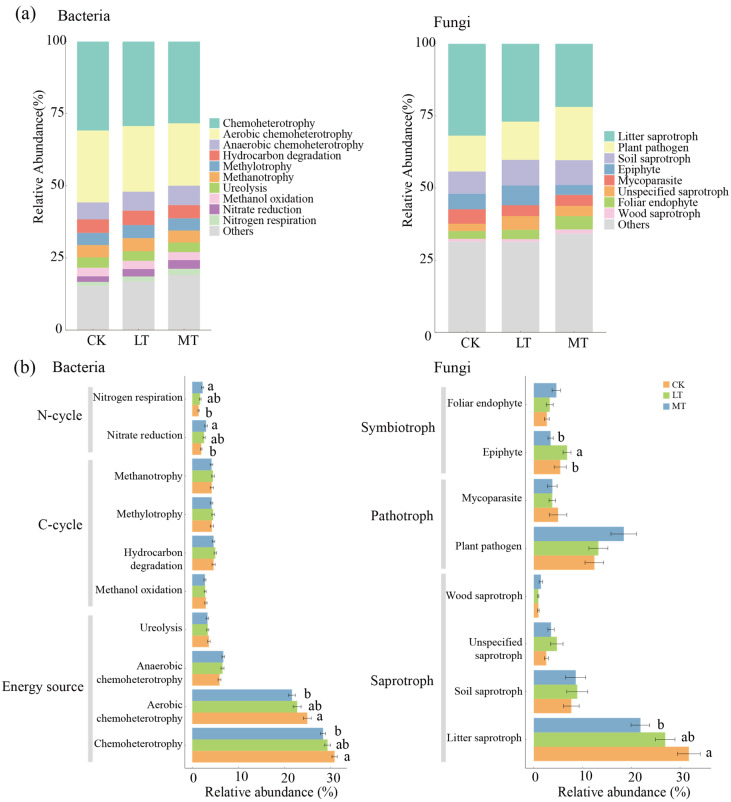
Functional prediction of endophytic fungal and bacterial communities across different thinning intensities. (**a**) Functional composition of endophytic bacterial and fungal communities. (**b**) Differences in functional composition of endophytic bacterial and fungal communities. CK: control check; LT: light thinning; MT: moderate thinning. Values are means ± SE for panels (*n* = 40). Different letters represent significant differences (*p* < 0.05).

**Figure 6 jof-10-00470-f006:**
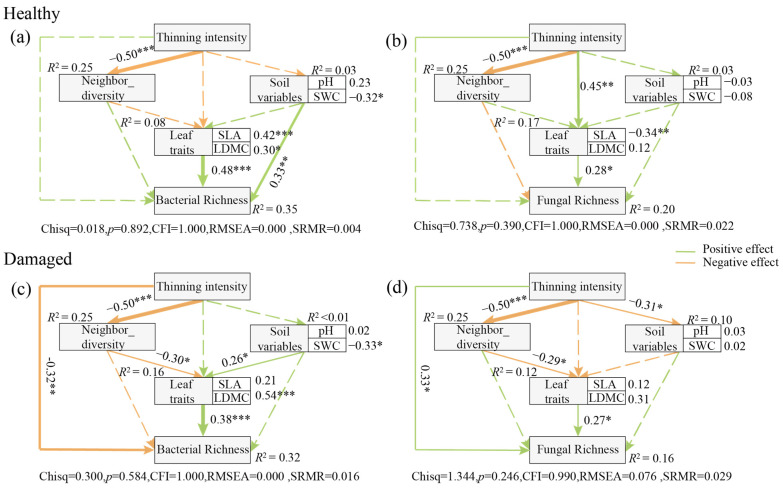
Structural equation models for evaluating factors closely associated with the richness of endophytic bacterial and fungal communities in leaves under different thinning intensities. (**a**) Endophytic bacteria in healthy leaves. (**b**) Endophytic fungi in healthy leaves. (**c**) Endophytic bacteria in damaged leaves. (**d**) Endophytic fungi in damaged leaves. Neighbor_diversity: neighbor tree richness; SLA: specific leaf area; LDMC: leaf dry matter content; pH: soil pH; SWC: soil moisture content. The green arrows indicate a positive correlation, and the orange arrows indicate a negative correlation. The solid lines indicate a significant relationship (*p* < 0.05), and the dashed arrows indicate an insignificant relationship (*p* > 0.05). The asterisks showed the *p*-value significance level—* *p* < 0.05, ** *p* < 0.01, *** *p* < 0.001.

**Table 1 jof-10-00470-t001:** Effects of different thinning intensity, tree species, damage degree, and their interactions on alpha diversity of endophytic fungi and endophytic bacteria in leaves.

Factor	Bacteria				Fungi			
	Richness		Shannon Index		Richness		Shannon Index	
	Chisq	*p*	Chisq	*p*	Chisq	*p*	Chisq	*p*
Thinning intensity (T)	4.40	0.111	3.11	0.21	14.8	<0.001	10.2	0.006
Species (S)	80.2	<0.001	26.3	<0.001	14.7	0.005	14.3	0.006
Damaged (D)	19.1	<0.001	21.5	<0.001	1.05	0.305	1.14	0.286
T × S	6.49	0.592	3.28	0.915	13.0	0.112	13.6	0.092.
T × D	0.21	0.899	0.32	0.852	0.69	0.71	0.27	0.875
S × D	20.8	<0.001	13.9	<0.001	5.13	0.274	3.80	0.434
T × S × D	4.96	0.762	6.26	0.617	2.75	0.949	7.89	0.444

**Table 2 jof-10-00470-t002:** Topological information of endophytic fungi and bacteria networks across different thinning intensities.

	Thinning Intensity	Edges	Nodes	Edge Density	Average Degree	Relative Modularity
Bacteria	CK	3286	470	0.030	13.983	1.761
	LT	1439	388	0.019	7.418	1.232
	MT	2574	448	0.026	11.491	1.48
Fungi	CK	17,922	1134	0.028	31.609	6.761
	LT	21,922	1264	0.027	34.687	7.198
	MT	23,597	1271	0.029	37.131	7.823

Note: relative modularity refers to the extent measures to which a network is compartmentalized into different modules. CK: control check; LT: light thinning; MT: moderate thinning.

**Table 3 jof-10-00470-t003:** Response of leaf traits, soil variables, and neighbor tree diversity to thinning intensity.

Factor	Thinning Intensity		
	CK	LT	MT
Ca (g/kg)	2.33 ± 0.19 a	2.14 ± 0.18 a	2.26 ± 0.19 a
Cu (mg/kg)	6.49 ± 0.32 a	6.39 ± 0.42 a	6.06 ± 0.39 a
K (g/kg)	9.84 ± 0.63 a	9.69 ± 0.55 a	9.03 ± 0.54 a
Mg (g/kg)	2.31 ± 0.15 a	2.21 ± 0.12 a	2.31 ± 0.14 a
**Zn (mg/kg)**	37.30 ± 2.43 **a**	33.07 ± 3.12 **ab**	24.84 ± 2.35 **b**
**P (g/kg)**	2.3 ± 0.08 **a**	2.06 ± 0.06 **b**	2.09 ± 0.07 **ab**
C (g/kg)	461.13 ± 2.99 a	461 ± 4.78 a	462.07 ± 5.17 a
N (g/kg)	24.90 ± 1.11 a	22.14 ± 0.8 a	22.56 ± 0.88 a
C/N	20.48 ± 1.21 a	22.41 ± 1.19 a	22.28 ± 1.24 a
**SLA (cm^2^/g)**	204.27 ± 16.88 **a**	155.06 ± 13.64 **b**	160.02 ± 11.88 **ab**
LA (cm^2^)	27.54 ± 2.53 a	23.17 ± 2.03 a	25.49 ± 2.04 a
**LDMC (g/g)**	0.32 ± 0.02 **ab**	0.31 ± 0.01 **b**	0.35 ± 0.01 **a**
LL (cm)	10.11 ± 0.37 a	9.67 ± 0.29 a	9.76 ± 0.33 a
LW (cm)	4.80 ± 0.41 a	4.44 ± 0.39 a	4.74 ± 0.41 a
LT (mm)	0.17 ± 0.01 a	0.17 ± 0.01 a	0.16 ± 0.01 a
**Neighbor_diversity**	0.99 ± 0.08 **a**	0.65 ± 0.06 **b**	0.35 ± 0.08 **c**
**pH**	4.40 ± 0.07 **a**	4.29 ± 0.08 **a**	3.99 ± 0.08 **b**
SWC (%)	47.05 ± 0.88 a	45.22 ± 1.25 a	45.57 ± 1.62 a

Note: Ca: total leaf calcium; Cu: total leaf copper; K: total leaf potassium; Mg: total leaf magnesium; Zn: total leaf zinc; P: total leaf phosphorus; C: total leaf carbon; N: total leaf nitrogen; C/N: carbon/nitrogen ratio; SLA: specific leaf area; LA: leaf area; LDMC: leaf dry matter content; LL: leaf length; LW: leaf width; LT: leaf thickness; Neighbor_diversity: neighbor tree richness; pH: soil pH; SWC: soil moisture content. CK: control check; LT: light thinning; MT: moderate thinning. Values are means ± SE for panels (*n* = 40). Different letters represent significant differences (*p* < 0.05). Bold the indicators with significant differences under different thinning intensities and bold the letters.

## Data Availability

The microbial raw DNA sequences have been deposited into the NCBI database, with the accession numbers SRR29007114-SRR29007233 and SRR29019700-SRR29019819.

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
