# Peer review of "Contrasting Patterns of Fungal and Bacterial Endophytes Inhabiting Temperate Tree Leaves in Response to Thinning"

_jof, 2024, doi:10.3390/jof10070470_

Round 1

Reviewer 1 Report

I have read the manuscript from the perspective of plant pathology, and less from the perspective of ecology. Consequently, it would have been useful for me to read more about the incidence reports of the key taxa identified in quadrants II and IV (Fig. 4) in any of the five host species studied.

Although reference is made to Methylobacterium (L460) and Erythrobasidium (L463), it is not specified if they have ever been reported in the host species (genus or family) analyzed in this study. Additionally, culturable endophytes, regardless of their ecological role in nature, still remain important for the development of biological control agents. I suggest, if the authors consider it pertinent, to include some lines for readers interested in plant protection.

General

- I recommend that supplemental figures also have a title, as well as what the codes in the tables (Tab 2: RM, CK, etc.) and figures (Fig. 6) mean. This way, they would be self-explanatory, without needing to navigate to M&M's to remember what they mean.

- Homogenize the names of the variables in the Tab. 3 and Fig. 6.

Reviewer 2 Report

Dear Colleagues.
What could be responsible for the change in species diversity of endophytic microorganisms? Have you measured the illumination, humidity
and wind speed in the areas?

You have explored different thinning intensities. Have you tried to study areas with higher intensity, for example > 50%?

Line 109
“0% (control, CK) Why "CK" but not just "control"?, 15% (light thinning, LT) and 30% (moderate thinning, MT) [33]”

Line 337
Table 1. Effects of different thinning intensity, tree species, damage degree, and their interactions 224 on Alpha diversity of endophytic fungi and endophytic bacteria in leaves.

It is not very clear from the table how exactly the intensity of thinning (Line 109 says) and indicators are related?

Line 338.
“The dominant communities include litter saprotrophs, plant pathogens, and soil saprotrophs, accounting for more than 50% of the communities (Fig. 5a).”
The same information is repeated twice. It is better to remove the pathotroph by replacing it with a plant pathogen.

Round 2

Reviewer 2 Report

Dear colleagues, thank you for your answer. With the changes made, the article has become clearer.

With the changes made, the article can be published